# Combined Therapy of Locally Advanced Oesophageal and Gastro–Oesophageal Junction Adenocarcinomas: State of the Art and Aspects of Predictive Factors

**DOI:** 10.3390/cancers13184591

**Published:** 2021-09-13

**Authors:** Milan Vošmik, Jindřich Kopecký, Stanislav John, Ondřej Kubeček, Petr Lochman, Aml Mustafa Banni, Libor Hruška, Igor Sirák

**Affiliations:** 1Department of Oncology and Radiotherapy, Charles University, Faculty of Medicine in Hradec Králové and University Hospital Hradec Králové, 500 05 Hradec Králové, Czech Republic; jindrich.kopecky@fnhk.cz (J.K.); stanislav.john@fnhk.cz (S.J.); ondrej.kubecek@fnhk.cz (O.K.); amlmustafa.banni@fnhk.cz (A.M.B.); libor.hruska@fnhk.cz (L.H.); igor.sirak@fnhk.cz (I.S.); 2Department of Surgery, Charles University, Faculty of Medicine in Hradec Králové and University Hospital Hradec Králové, 500 05 Hradec Králové, Czech Republic; petr.lochman@fnhk.cz; 3Department of Field Surgery, Faculty of Military Health Sciences, University of Defence, 500 05 Hradec Králové, Czech Republic

**Keywords:** oesophagus, gastro-oesophageal junction, adenocarcinoma, perioperative chemotherapy, preoperative chemoradiotherapy

## Abstract

**Simple Summary:**

The optimal treatment strategy for locally advanced distal oesophageal and gastrooesophageal junction (GOJ) adenocarcinoma is currently not clear. Surgery as a main curative modality is usually combined with either preoperative chemoradiotherapy or perioperative chemotherapy. The aim of the review is to provide an overview of current treatment options in locally advanced oesophageal and GOJ adenocarcinomas based on the latest evidence, including the possible potential of predictive biomarkers in optimizing treatment.

**Abstract:**

The following main treatment approaches are currently used in locally advanced adenocarcinomas of the oesophagus and gastrooesophageal junction (GOJ): preoperative chemoradiotherapy and surgery, and perioperative chemotherapy and surgery. While preoperative chemoradiotherapy is used primarily in oesophageal tumours, perioperative chemotherapy is the treatment of choice in Western countries for gastric cancer. The optimal treatment strategy for GOJ adenocarcinoma is still not clear. In comparison to other malignancies, biomarkers are used as predictive factors in distal oesophageal and GOJ adenocarcinomas in a very limited way, and moreover, only in metastatic stages (e.g., HER2 status, or microsatellite instability status). The aim of the article is to provide an overview of current treatment options in locally advanced adenocarcinomas of oesophagus and GOJ based on the latest evidence, including the possible potential of predictive biomarkers in optimizing treatment.

## 1. Introduction

The epidemiology of gastroesophageal tumours has been changing over the last decades in Europe and North America. The incidence of gastric cancer declined and the decrease is explained mainly by the clearance of Helicobacter Pylori [1]. The incidence of oesophageal tumours seems to be stable, but the changes are obvious when considering the histologic subtype. While the incidence of oesophageal squamous cell carcinoma decreases, the number of patients with adenocarcinoma of oesophagus and gastro-oesophageal junction (GOJ) is on the rise making it the dominant histological cancer type [2,3]. The following are the main risk factors for oesophageal and GOJ adenocarcinomas: gastro-oesophageal reflux disease, Barrett’s oesophagus, obesity, cigarette smoking, high processed meat intake, and low intake of vegetables and fruit [4,5].

Although surgery is still the main curative modality, combined therapy approaches have become the standard of care (SOC) in locally advanced oesophageal and gastric cancers. Preoperative chemoradiotherapy and surgery is commonly used in locally advanced resectable oesophageal tumour, whereas surgery combined with perioperative chemotherapy, or surgery followed by postoperative chemoradiotherapy, has been adopted as the SOC in gastric cancer in Western countries. With regards to locally advanced adenocarcinomas of the distal oesophagus and GOJ, the optimal treatment strategy is much less clear. These tumours were included in large randomized clinical trials that have defined standard practice in oesophageal carcinomas as well as in “practice changing” clinical trials in gastric cancer. Therefore, we can treat oesophageal and GOJ adenocarcinomas according to guidelines for oesophageal tumours using preoperative chemoradiotherapy, but also as primary tumour of the stomach with perioperative chemotherapy. Furthermore, the results of some randomized clinical trials with innovative drugs, including immune-check-point inhibitors, that have already changed treatment standards and further clinical research, have recently been published.

Clinical research is focused on refining practice guidelines for diagnosis, classification, and personalized therapy. The aim of this review is to provide an overview of current treatment options in locally advanced adenocarcinomas of oesophagus and GOJ based on the latest evidence, including the possible potential of predictive biomarkers in optimizing treatment.

## 2. Methods

An electronic literature search was conducted in the PubMed database for English articles published up to July 2021. Search terms used were: “esophageal neoplasm”, “esophageal neoplasms”, “esophageal cancer”, “cancer of the esophagus”, “cancer of esophagus”, “esophagus cancer”, “esophageal cancer”, “oesophageal neoplasm”, “oesophageal neoplasms”, “oesophageal cancer”, “cancer of the oesophagus”, “cancer of oesophagus”, “oesophagus cancer”, “oesophageal cancer”, “gastroesophageal junction”, “gastrooesophageal junction”, “gastro-esophageal junction”, “gastro-oesophageal junction”, “randomised“, “neoadjuvant“, “preoperative“, “perioperative“ “adjuvant“, “postoperative”, “adenocarcinoma“, “predictive biomarker“, and “predictive factor“. The papers and clinical trials were selected by a relevancy. Significant abstracts published at important congresses were added to the review.

## 3. Anatomical Terminology

To unify the terminology, Siewert proposed a classification that defined three types of GOJ tumours [6,7] and this classification is now used by most authors. Siewert type I tumours are located in the distal oesophagus and the centre of the tumour is more than 1 cm and up to 5 cm above the anatomical junction. Type II tumours originate in the region of the anatomical junction (cardia) and the centre of the tumour is up to 1 cm above and up to 2 cm below the junction. Type III tumours are primarily subcardial tumours and their centre is more than 2 cm and up to 5 cm below the junction. Unfortunately, not all authors reported data using the Siewert classification and therefore, the comparison of clinical trials results is difficult.

The development of the TNM classification as a basic tool for defining the clinical stage of tumours, reflects uncertainties how to classify GOJ tumours as well. The sixth edition from 2002 simply classifies GOJ tumours as gastric carcinomas in accordance with the ICD classification under the code C16.0 [8]. According to the seventh edition, all GOJ tumours with the centre up to 5 cm below the anatomical junction and at the same time growing into the oesophagus, were to be classified as oesophageal tumours, whereas tumours with the centre more than 5 cm from the junction or without growing into the oesophagus, as gastric carcinomas [9]. The currently valid eighth edition mentions the Siewert classification, however, the terminological classification does not correspond to the distances defined by Siewert. According to the eighth edition, tumours affecting the GOJ, in which the centre is up to 2 cm proximal to the cardia, are classified as oesophageal carcinomas. On the contrary, tumours whose centre is more than 2 cm distal to the GOJ are divided into stages according to the classification for gastric tumours, even if the GOJ is affected. GOJ tumours (C16.0) are considered to be tumours with a centre up to 2 cm proximal and up to 2 cm distal to the anatomical junction [10].

## 4. Surgical Treatment of Locally Advanced Oesophageal and GOJ Adenocarcinomas

The aim of the surgical treatment of locally advanced oesophageal and GOJ adenocarcinomas is to achieve an R0 resection. The type of surgery depends predominantly on primary tumour location and availability of potential conduit for reconstruction. The two most common types of oesophagectomy are transthoracic (Ivor-Lewis and McKeown oesophagectomy) and transhiatal (Orringer oesophagectomy), mostly with gastric conduit used for reconstruction. Transthoracic approach by Ivor-Lewis with intrathoracic anastomosis in its open or miniinvasive form is suitable for tumours located in middle and distal part of thoracic oesophagus, and for GOJ tumours, as well. In the case of an upper thoracic oesophageal tumour, McKeown oesophagectomy with cervical anastomosis is the method of choice. Tumours located in distal oesophagus and GOJ can be managed by transhiatal approach combined with an incision on the neck and cervical anastomosis.

Surgical management of GOJ tumours depends on the extent of the tumour to the oesophagus, possibility to reach negative proximal margin and to perform an adequate lymphadenectomy. Siewert type I tumours should be managed by oesophagectomy (usually transthoracic, transhiatal only if oesophageal involvement is <4 cm) with gastric conduit replacement. Siewert type III tumours are mostly indicated for total gastrectomy with D2 lymphadenectomy and reconstruction with oesophago-jejunal anastomosis by Roux-Y. The surgical management of true cardia primary tumours (Siewert type II) is controversial. Subtotal oesophagectomy and extended total gastrectomy with distal oesophagectomy are equally preferred options.

The extent of gastric surgery is influenced by a required extent of lymphadenectomy, and therefore, the surgical management of lymph nodes of GOJ tumours is largely discussed. Kurokawa et al. published recently results of GOJ tumours lymph node metastases mapping and based on these results presented the following recommendation. In patients with Siewert type II tumours, proximal gastrectomy with distal oesophagectomy is accepted because the incidence of metastases in the distal perigastric nodes is low. Dissection of upper mediastinal lymph nodes (station 106recR) should be performed in patients with esophageal involvement of more than 4.0 cm, and lower mediastinal lymph nodes (station 110) should be removed in cases with esophageal involvement over 2.0 cm [11].

## 5. Combined Treatment Options in Locally Advanced Oesophageal and GOJ Adenocarcinomas

As mentioned above, the combination of surgery with non-surgical modalities (chemotherapy +/− radiotherapy) is SOC in locally advanced resectable adenocarcinomas of oesophagus and GOJ. According to the European Society for Medical Oncology (ESMO), National Comprehensive Cancer Network (NCCN), and American Society of Clinical Oncology (ASCO) guidelines, the following treatment approaches combined with surgery can be considered: preoperative chemoradiotherapy, postoperative chemoradiotherapy, perioperative chemotherapy, and postoperative chemotherapy [12,13,14], although other combined strategies were tested in randomised trials as well. Recently, targeted therapy and immune checkpoint inhibitors are increasingly used in a number of malignancies and clinical trials verifying the effectiveness of these innovative drugs in locally advanced oesophageal and GOJ adenocarcinomas are being conducted. The first results of some of these studies have already been published and one of these novel approaches, adjuvant immunotherapy with anti-PD-1 antibody nivolumab in locally advanced oesophageal and GOJ tumours after previous preoperative chemoradiotherapy and R0 resection, has been already adopted as a SOC [14,15]. A list of the main randomised clinical trials influencing clinical practice in locally advanced oesophageal and gastro-oesophageal junction adenocarcinoma is in Table 1.

## 6. Preoperative Chemoradiotherapy

As a preoperative therapy, concurrent chemoradiotherapy (CRT) has been used historically mainly in oesophageal tumours. The reason is that chemoradiotherapy has been confirmed in clinical trials as a definitive treatment [16,17]. Clinical trials and meta-analyses then demonstrated a benefit of preoperative chemoradiotherapy compared to surgery alone [18,19,20]. However, most of these trials enrolled together patients with both basic histology types, squamous cell carcinomas and adenocarcinomas. In adenocarcinomas alone, the benefit was considered to be ambiguous.

Several following clinical trials enrolled patients with a significant number of adenocarcinomas. Walsh et al. conducted a clinical trial in the 1990s that compared preoperative chemoradiotherapy (RT 40 Gy in 15 fractions and two cycles of cisplatin and 5-fluorouracil) and surgery to surgery alone only in oesophageal adenocarcinomas. In the multimodality group, 25% of complete pathological responses were observed, the median overall survival was 17 months compared to 12 months in patients treated with surgery alone (*p* = 0.002). Three-year overall survival was 32% in the multimodality arm compared to 6% in the surgery alone arm (*p* = 0.01) [21,22].

A single-institution trial at the University of Michigan randomised 100 patients with oesophageal cancer (adenocarcinomas in 75%) to preoperative chemotherapy (cisplatin, 5-fluorouracil and vinblastin) and radiotherapy (45 Gy; 1.5 Gy twice a day) and surgery versus surgery alone. The three-year cause-specific survival rate was 30 % and 16%, respectively, but the difference was not statistically significant (*p* = 0.15) [23].

TROG/AGITG intergroup trial randomised 256 patients (with adenocarcinoma or mixed histology in 63% of tumours) to preoperative chemoradiotherapy (RT 35 Gy in 15 fractions and concurrent cisplatin and 5-fluorouracil) and surgery or to surgery alone. This trial, in contrast with the study of Walsh et al., did not find a statistically significant survival benefit. In particular, the difference was negligible in non-squamous histology (overall survival *p* = 0.81, HR 1.04; 95% CI 0.74–1.48) [24].

The phase III, randomised trial CALGB 9781 was designed to compare preoperative chemoradiotherapy (RT 55.8 Gy in 6 weeks and concurrent cisplatin and 5-fluorouracil) and surgery versus surgery alone in 475 patients. Unfortunately, the trial was closed due to poor enrolment and therefore, only 56 patients were evaluated (adenocarcinomas were 77% of tumours). Despite this fact, an intent-to-treat analysis showed a median survival of 4.5 versus 1.8 years in favour of trimodality therapy (*p* = 0.002) and five-year survival reached 39% in the trimodality arm compared to 16% in the surgery alone arm [25].

The clear benefit of preoperative chemoradiotherapy and surgery in a subset of patients with distal oesophageal adenocarcinoma, including GOJ, was unequivocally confirmed in the largest randomized trial, CROSS [26,27,28]. This trial compared preoperative chemoradiotherapy and surgery with surgery alone in oesophageal and GOJ tumours and analysed treatment outcomes for both histological types separately. Preoperative chemoradiotherapy consisted of radiation (41.4 Gy in 23 fractions) and concurrent chemotherapy with paclitaxel and carboplatin. The total number of patients in the trial was 366 patients and the proportion of adenocarcinomas in both arms was 75%. The majority of tumours were located in the oesophagus; the GOJ was a primary site in 24 %. The median overall survival was 48.6 months in the combined modality arm compared to 24.0 months in the surgery alone arm (*p* = 0.003) and the median survival in the adenocarcinoma subgroup was 43.2 months compared to 27.1 months, respectively (*p* = 0.038) [27]. The ten-year overall survival rates were in the whole group and in the adenocarcinoma subgroup 38% compared to 25% (*p* = 0.004), and 36% versus 26%, respectively (*p* = 0.061) [28]. It is not without interest that in addition to reducing the risk of locoregional progression (22% versus 38%; *p* < 0.0001), the risk of distal progression was reduced in the combination arm as well (39% versus 48%; *p* = 0.004) [27].

## 7. Postoperative Chemoradiotherapy and Chemotherapy

The benefit of postoperative chemoradiotherapy in locally advanced gastric or GOJ tumours was demonstrated in the randomised SWOG 9008/INT 0116 trial. This trial compared surgery followed by postoperative chemoradiotherapy (six cycles of bolus 5-fluorouracil and leucovorin in a bolus 5-day regimen and radiotherapy at a dose of 45 Gy in 25 fractions) and surgery alone. The trial randomised 556 patients and the primary tumour in the gastroesophageal junction was present in approximately 20 percent of patients. Postoperative chemoradiotherapy was associated with a lower risk of relapse and prolongation of overall survival (median 36 months versus 27 months, *p* = 0.005) [29,30]. However, 54% of enrolled patients in the whole group had <D1 lymph node dissection and therefore, the quality of surgery is not considered to be optimal in this trial.

CALGB 80101 was a randomised, phase III, clinical trial that compared two regimens of postoperative chemotherapy combined with concurrent chemoradiotherapy. The trial primarily studied patients with gastric tumours but GOJ adenocarcinomas represented 22% of all tumours. The same regimen of chemoradiotherapy was applied in both arms (RT: 45 Gy in 5 weeks and concurrent continual infusion of 5-fluorouracil). One cycle of chemotherapy before and two cycles after the chemoradiotherapy were administered. Chemotherapy consisting of 5-fluorouracil and leukovorin in a bolus 5-day regimen was applied in the standard arm and a combination of epirubicin, cisplatin and 5-fluorouracil (ECF regimen) was administered in the experimental arm. However, this trial did not demonstrate any survival advantage of combining of postoperative chemoradiotherapy with ECF regimen compared to 5-fluorouracil and leukovorin [31].

## 8. Perioperative Chemotherapy

The French randomised FNCLCC/FFCD trial compared surgery combined with perioperative chemotherapy (cisplatin and 5-fluorouracil) with surgery alone. Of the 224 patients enrolled in the study, 11% had tumours of the distal oesophagus and 64% had tumours of GOJ. The trial showed an improvement in five-year overall survival in the perioperative chemotherapy arm (34% versus 19%; *p* = 0.003) [32].

The British MAGIC trial enrolled a total of 503 patients with gastric, GOJ or distal oesophageal adenocarcinoma into an experimental arm with CHT consisting of epirubicin, cisplatin and 5-fluorouracil or capecitabine (ECF or ECX), administered as three cycles preoperatively and three cycles postoperatively or into the standard arm with surgery alone. The majority of patients had gastric cancer. The primary tumour site was distal oesophagus in 11.5% and GOJ in 14.5%, respectively. Overall survival was improved in the perioperative CHT arm (five-year overall survival 36% versus 23%; *p* = 0.009). However, only 42% of patients in the perioperative-chemotherapy arm completed all protocol treatment and 34% of patients who completed preoperative chemotherapy and surgery did not begin postoperative chemotherapy. The reasons were either early disease progression, patient request or postoperative complications [33].

The randomised, phase 2/3, FLOT4-AIO trial compared the ECF or ECX regimen used in the MAGIC study with the FLOT regimen (combination of docetaxel, oxaliplatin, 5-fluorouracil, and leucovorin). The 716 patients, of which 56% had tumours of distal oesophagus (Siewert I; 24%) or GOJ (Siewert II–III; 32%), were enrolled in this study. The overall survival in the FLOT arm was significantly longer than in the ECF/ECX arm (median 50 months versus 36 months; *p* = 0.012), with comparable toxicity of both regimens [34]. The FLOT regimen has thus become the current standard for perioperative CHT in these tumours.

## 9. Preoperative Chemotherapy

The United Kingdom Medical Research Council (UK MRC) Oesophageal Cancer Working group conducted a randomised trial OEO2 that compared preoperative chemotherapy (two cycles of cisplatin and 5-fluorouracil) and surgery with surgery alone. A total of 802 patients with oesophageal cancer, including the cardia, were enrolled. Histology of adenocarcinoma was reported in 67% of tumours. This study demonstrated a survival benefit in the preoperative chemotherapy arm with a five-year overall survival of 23.0% in the preoperative chemotherapy arm compared to 17.1% in the surgery alone arm (*p* = 0.03) [35].

In the RTOG 89-11/INT 113 trial, 440 patients with oesophageal or GOJ cancer were randomised to preoperative chemotherapy (three cycles of cisplatin and 5-fluorouracil) and surgery or surgery alone. Histology of adenocarcinoma was reported in 55% of tumours in the whole group. In contrast to OEO2, this trial did not demonstrate any survival benefit of preoperative chemotherapy and surgery compared to surgery alone [36].

Based on the results of OEO2 and MAGIC trials, the UK MRC conducted a randomised trial that compared two regimens of preoperative chemotherapy: two cycles of cisplatin and 5-fluorouracil with four cycles of ECX regimen. A total of 897 patients with adenocarcinoma of the distal oesophagus or GOJ (Siewert I–II) were enrolled, however, this trial did not find any difference in survival using the intensified regimen [37].

## 10. Postoperative Chemotherapy

Based on the ACTS-GC Group study and the CLASSIC study, adjuvant chemotherapy has become standard treatment in gastric cancer after gastrectomy and D2 lymphadenectomy, especially S1 or oxaliplatin and capecitabine (XELOX regimen) in Asian countries [38,39]. The CLASSIC trial included 24 patients with adenocarcinoma of GOJ. Although it was a very small number (only 2.3%), NCCN guidelines extrapolated the results and endorsed adjuvant XELOX or FOLFOX as a treatment option for patients with GOJ and distal oesophagus tumours who received a preoperative treatment [14].

## 11. Combination of Perioperative Chemotherapy and Postoperative Chemoradiotherapy

The phase III CRITICS trial, conducted in Netherlands, Sweden, and Denmark, randomised 788 patients with gastric or GOJ adenocarcinomas to perioperative chemotherapy (regimen ECX or EOX; epirubicin, cisplatin or oxaliplatina, and capecitabine; three cycles preoperatively and three cycles postoperatively) and surgery or preoperative chemotherapy (regimen ECX or EOX; three cycles preoperatively) and surgery, followed by postoperative chemoradiotherapy (RT: 45 Gy in 25 fractions, concurrent chemotherapy: capecitabine and cisplatin). This study included 17% of patients with tumours of GOJ. Postoperative chemoradiotherapy did not improve overall survival compared with postoperative chemotherapy [40].

## 12. Comparison of Preoperative or Perioperative Chemotherapy and Preoperative Chemoradiotherapy

To date, the only randomised phase III trial that compared preoperative chemotherapy and surgery with trimodality therapy in patients with GOJ adenocarcinomas (Siewert I–III) only, is the German POET study. The trial evaluated preoperative CHT, combination of 5-fluorouracil, leucovorin and cisplatin for 17 weeks, and the same preoperative chemotherapy for 14 weeks, followed by radiotherapy at a dose of 30 Gy and concurrent in combination with cisplatin and etoposide for 3 weeks. Unfortunately, this German study was terminated prematurely due to slow patient recruitment (*n* = 126). Although a required level of statistical significance was not reached in the survival analysis, the results favour the trimodality therapy. Three-year and five-year overall survival was 26.1% and 24.4% in the chemotherapy arm, and 46.7% and 39.5% in the chemoradiotherapy arm, respectively (*p* = 0.055). The only statistically significant benefit was a probability of a complete response of 1.9% versus 14.3% (*p* = 0.03) [41,42].

Recently, preliminary results from interim analysis of the Neo-AEGIS trial were presented at the virtual ASCO meeting. This trial is being conducted in Ireland, the United Kingdom, Denmark, France, and Sweden. A total number of 362 evaluable patients with locally advanced adenocarcinoma of the oesophagus or GOJ (Siewert I–III) were randomly assigned to one arm with preoperative chemoradiotherapy according to the CROSS trial or the other arm with perioperative chemotherapy (ECF/ECX/EOF/EOX until 2018 or FLOT since 2019). The interim analysis did not note any significant difference in overall survival. Three-year estimated survival probability was 56% and 57%, respectively (HR 1.02; 95% CI 0.74–1.42). The toxicity was higher in the perioperative chemotherapy arm (mainly neutropenia, diarrhoea, and vomiting). The R0 resections, complete responses and nodal downstaging were more frequent in the CROSS arm, nevertheless the risk of complications associated with radiotherapy was not significantly higher. The final assessment of this trial is expected in 2022 [43].

A similar trial comparing preoperative chemoradiotherapy (CROSS protocol) with perioperative chemotherapy (FLOT protocol) is currently being conducted in Germany. The inclusion criteria allow the enrolment of patients with locally advanced adenocarcinomas of GOJ Siewert type I, and Siewert II–II in case of oesophageal infiltration [44].

The benefit of preoperative CHRT added to perioperative chemotherapy over perioperative chemotherapy without radiotherapy is currently being evaluated in gastric and GOJ tumours in a randomized phase III study TOPGEAR [45]. In the meantime, an interim analysis of the first 120 patients was published and the results confirmed that preoperative chemoradiotherapy can be delivered without a significant increase in treatment toxicity and surgical morbidity [46].

## 13. Targeted Therapy and Immune Check-Point Therapy

### 13.1. Anti-VEGF Therapy

The UK MRC phase II–III trial ST03; the follow-on study for the MAGIC trial, randomized 1063 patients with gastric, GOJ, or distal oesophageal adenocarcinoma to perioperative chemotherapy ECX or the same perioperative chemotherapy combined with anti-VEGF antibody bevacizumab. The primary tumour site was the distal oesophagus (14%), GOJ Siewert type I (12%), Siewert type II (19%), Siewert type III (20%), and stomach (36%). This trial did not confirm that the addition of bevacizumab to perioperative chemotherapy for patients with resectable tumours is associated with a statistically significant overall survival benefit. The three-year overall survival was 50.3% in the chemotherapy alone group compared to 48.1% in the chemotherapy plus bevacizumab group (*p* = 0.36). The important reported finding was that the addition of bevacizumab to chemotherapy was associated with a higher risk of postoperative complications, mainly in wound healing. Higher rate of postoperative anastomotic leak in patients who underwent oesophagogastrectomy led to the premature end of recruitment of patients with lower oesophageal or junctional tumours planned for an oesophagogastric resection [47].

### 13.2. Anti-HER2 Therapy

The tyrosin kinase receptor HER2 (also known as Erb-B2) is a member of the epidermal growth factor receptor (EGFR) family. It is encoded by protooncogene *HER2* (also known as *ERBB2*). *HER2* gene amplification leads to HER2 overexpression and the finding of *HER2* amplification and/or HER2 overexpression (called HER2-positivity) has been expected to be associated with a beneficial response to anti-HER2 therapy. Based on the results of the TOGA trial, the anti-HER2 antibody trastuzumab in combination with chemotherapy became SOC as the first line of palliative systemic therapy in HER2-positive advanced/metastatic gastro-oesophageal adenocarcinomas [48]. Therefore, anti-HER2 treatment is tested in locally advanced HER2-positive tumours as a component of preoperative or postoperative treatment. Phase III trial RTOG-1010 randomised 203 patients with HER2-positive adenocarcinoma involving the mid, distal oesophagus, or GOJ and up to 5 cm of the stomach, to chemotherapy (paclitaxel, carboplatin) and radiotherapy (50.4 Gy in 28 fractions) followed by surgery, or the same treatment with weekly trastuzumab prior to surgery and every 3 weeks for 39 weeks after the surgery. According to the preliminary results, the addition of trastuzumab to trimodality therapy does not improve disease-free survival, nor does it worsen the toxicity [49].

The German AIO “PETRARCA” trial was planned to compare the perioperative FLOT regimen alone (standard arm) or in combination with trastuzumab and pertuzumab (experimental arm) in locally advanced gastric and GOJ HER-positive tumours [50]. The trial was closed prematurely after the JACOB trial did not demonstrate benefits of combining trastuzumab and pertuzumab with chemotherapy in comparison to trastuzumab and chemotherapy in the palliative setting in metastatic gastric and GOJ HER-positive tumours [51]. Therefore, only 81 patients were enrolled and evaluated. In the experimental arm, there was a higher rate of complete pathological response (12% versus 35%, respectively, *p* = 0.02) and pathological lymph node negativity (39% versus 68%). R0 resections and postoperative morbidity were comparable. Two-year disease-free survival and overall survival were 54% versus 70%, and 77% versus 84%, in the standard arm and experimental arm, respectively [50].

The randomised, phase II trial EORTC-1203 “INNOVATION” is a currently ongoing clinical trial comparing perioperative chemotherapy (FOLFOX, CAPOX, cisplatin with 5-fluorouracil or capecitabine, or FLOT regimen as a standard arm) with the same perioperative chemotherapy regimen combined with trastuzumab (experimental arm 1) or trastuzumab and pertuzumab (experimental arm 2) in locally advanced gastric or GOJ adenocarcinoma (Siewert I–III) [52].

### 13.3. Immune Check-Point Therapy

Global, double-blind, placebo-control phase III trial CheckMate 577 randomised 794 patients with resected stage II or III oesophageal or gastroesophageal junction cancer with residual pathological disease after preoperative chemoradiotherapy. Adenocarcinomas represented 71% of tumours. The patients were randomly assigned in a 2:1 ratio to receive nivolumab or placebo for a maximum period of 1 year. The first analysis demonstrated a disease-free survival benefit in patients who received nivolumab; the median disease-free survival was 22.4 months in the nivolumab arm compared to 11.0 months in the placebo arm (*p* < 0.001). Hazard ratios for disease recurrence or death favoured nivolumab across multiple prespecified subgroups, including PD-L1 expression or histological type. Disease-free survival in the adenocarcinoma subgroup was 19.4 months in the nivolumab arm versus 11.1 months in the placebo arm (HR 0.75; 95% CI 0.59–0.96). The benefit of nivolumab however, is questionable in the GOJ tumours subgroup (HR 0.87; 95% CI 0.63–1.21), although this result is not influenced by histological type (the adenocarcinoma subgroup profited from nivolumab as mentioned above) or by number of patients in the subgroup (the number of patients with GOJ tumour was 332) [15].

The DANTE study is a German phase II randomised trial that evaluates the effect of anti-PD-L1 antibody atezolizumab in combination with FLOT regimen compared to FLOT regimen alone as a perioperative treatment of locally advanced, potentially resectable gastric or GOJ adenocarcinoma [53]. KEYNOTE 585 trial is a phase III, randomised, double-blind study comparing perioperative chemotherapy (cisplatin plus 5-fluorouracil or capecitabine, or FLOT) and the same chemotherapy combined with anti-PD-1 antibody pembrolizumab. This study enrolls patients with locally advanced adenocarcinoma of the stomach or GOJ (Siewert type I–III) [54].

The EORTC VESTIGE trial is a randomised, phase II, trial enrolling patients with locally advanced adenocarcinomas of the stomach, GOJ (Siewert type I–III), or distal oesophagus after preoperative chemotherapy and surgery with a high risk of recurrence (ypN1-3 status or R1 resection). The patients are randomised postoperatively to the same chemotherapy as preoperatively (standard arm) or to immunotherapy with anti-CTLA-4 antibody ipilimumab plus anti-PD-1 antibody nivolumab for 1 year [55].

## 14. Predictive Biomarkers

Discoveries in the field of cancer molecular biology led to the accelerated development of new anticancer molecules targeting specific proteins that are involved in the growth and survival of cancer cells. Research has focused on the search for predictive markers, revealing a higher probability of treatment response to these innovative therapies. A molecular predictive biomarker may be the overexpression of a specific protein, amplification or mutation of a specific gene, etc. The development of predictive cancer diagnostics is linked to the treatment strategy called personalized medicine. In some malignancies, it is necessary to examine predictive markers before administering systemic therapy as is the case in palliative therapy in metastatic disease, but increasingly even in locally advanced stages, as a part of combined curative therapy.

The main studied molecular biomarkers are: HER2 overexpression/*HER2* gene amplification, PD-L1 expression, microsatellite instability (MSI) status/mismatch repair (MMR) status, and tumour mutation burden (TMB). HER2 overexpression/*HER2* gene amplification is a molecular predictive marker commonly used in distal oesophageal, GOJ, and gastric adenocarcinomas. HER2 overexpression or *HER2* gene amplification predicts a higher treatment response and survival benefit to anti-HER2 therapy in metastatic gastro-oesophageal adenocarcinomas [48]. Unfortunately, the usability of this marker is limited in locally advanced disease as there is still no evidence, as discussed above, that anti-HER2 treatment is associated with a survival benefit compared to standard treatment [49,50].

Similarly, PD-L1 expression is associated with a higher treatment response and survival benefit with some immune check-point inhibitors and malignancies [56]. Although there is data to use PD-L1 as a predictive factor for anti PD-1 therapy in the palliative setting [57,58], current evidence of immunotherapy in locally advanced adenocarcinomas of distal oesophagus and GOJ does not support the use of PD-L1 expression as a predictive marker since the benefit of nivolumab in the CheckMate 577 trial was independent of PD-L1 expression [15]. However, some further studies are being conducted, and PD-L1 expression therein may be shown as a predictor.

Other than PD-L1 expression, and other potential predictors of immune check-point inhibitor effectivity, it is particularly worthy to discuss microsatellite instability (MSI)/mismatch repair status (MMR), Epstein–Barr virus (EBV) status, and tumour mutation burden (TMB) [59,60]. MSI/MMR status especially seems to be a strong predictor of pembrolizumab therapy. Among 233 patients with MSI-high/MMR deficient noncolorectal tumours, including 24 patients with gastric carcinoma, treated with pembrolizumab in the palliative setting in the KEYNOTE-158 trial, the objective response rate was 34% and median overall survival was 23.5 months [59]. MSI-high status is reported in the range of 4–24% in adenocarcinomas of stomach and GOJ [61,62,63,64,65,66,67,68,69]. In general, MSI-high and/or MMR deficient tumours are associated with better prognosis compared to microsatellite stable (MSS) and MSI-low tumours [70,71,72,73]. Post hoc analyses of several trials [74,75] indicate that perioperative or postoperative chemotherapy improves the prognosis only in MSS/MSI-low status tumours. The meta-analysis of MAGIC, CLASSIC, ARTIST, and ITACA-S studies included 1556 patients with available MSI status, of which 7.8% of tumours were classified as MSI-high. Patients with MSI-low/MSS tumours had a survival benefit from combined therapy (five-year OS 62% vs. 53%; HR 0.75; 95% CI 0.60–0.94), whereas patients with MSI-high tumours did not (five-year OS 75% vs. 83%; HR 1.5; 95% CI 0.55–4.12) [76]. Although it appears that combined therapy has no benefit for MSI-high gastro-oesophageal adenocarcinomas, it is not possible to generalize this hypothesis to all chemotherapy regimens, including the current standard of perioperative chemotherapy, FLOT regimen, as e.g., taxane activity is probably independent of the mismatch repair system [77]. Nevertheless, clinical trials with immune check-point inhibitor therapy in MSI-high locally advanced gastro-oesophageal adenocarcinomas are desirable.

EBV is known to be the main pathogenic factor for nasopharyngeal carcinoma but surprisingly, it was found that EBV genome is present in a subset of gastric cancer [78]. EBV-positivity correlates with PD-L1 expression and increased TIL density [79]. Therefore, EBV positive gastric cancer is considered to be a candidate for immune check-point inhibitor therapy, as well. Kim et al. enrolled 61 patients, including seven patients with MSI-high tumours and six patients with EBV positive tumours, in a study with pembrolizumab therapy as a second-line or third-line treatment for metastatic gastric cancer. Overall, three patients (5%) achieved complete remission, 12 patients (20%) partial remission, and 20 patients (33%) had stable disease. Pembrolizumab was effective mainly in MSI-high tumours (three complete remissions and three partial remissions among seven patients), and in EBV positive tumours (partial remission was noted in all six cases) [80]. The most important mechanism of carcinogenesis in EBV-positive gastric cancer is probably hypermethylation of DNA leading to silencing of key tumour suppressor genes [81] and therefore, demethylating treatment strategies could be a therapeutic option in EBV-positive gastric cancer, including GOJ adenocarcinomas [82].

The *TP53* gene is probably the most frequently mutated gene in oesophagogastric adenocarcinomas. The frequency of *TP53* mutations in gastric tumours is 30–70% [83,84,85,86]. Mutations of *TP53* (point mutations or deletions) lead to the production of aberrant proteins that not only lose their tumor-suppressive functions but also frequently act as driver oncogenes that promote malignant progression, invasion, and metastasis [87]. Although no targeted therapy is currently available for *TP53* mutated tumours, this topic is the subject of research, which focuses on small molecular compounds, synthetic small peptides, CRISPR/Cas9 mediated genome editing, or small interference RNAs (RNAi) [88].

The loss of function of E-cadherin gene (*CDH1*) and E-cadherin deficiency is associated with the diffuse subtype according to the Laurén classification [89]. It is well known that the diffuse type of gastric cancer is associated with aggressive behavior and worse prognosis, and therefore, development of targeted E-cadherin therapy is also desirable [90].

A large number of other possible biomarkers are the subject of research. Many of these biomarkers belong to the category of biomarkers of tumour agnostic therapy, which means that they can be used as a biomarker in any malignant tumour. These biomarkers are used currently in advanced and metastatic tumours in the palliative setting, e.g., TRK inhibitors in NRTK fusion positive tumours [91,92].

In general, the increasing knowledge of molecular-biological characteristics of tumours and the awareness that these characteristics may influence the prognosis or even predict the treatment, has led to efforts to create classifications based on the molecular characteristics of tumours. In 2014, the Cancer Genome Atlas (TCGA) project published a proposal of tumour classification of gastric adenocarcinomas based on comprehensive molecular evaluation. This project divides gastric adenocarcinomas into four subtypes: Epstein–Barr virus (EBV) positive; microsatellite unstable cancer (MSI-high); genomically stable cancer (GS); and chromosomally unstable cancer (CIN). GS tumours have a higher probability of *CDH1* mutations (37%), whereas CIN tumours are often associated with *TP53* mutations (71%) [93]. The TCGA project also later published an analysis of oesophageal tumours [94], which showed that the vast majority of distal oesophagus and GOJ adenocarcinomas are CIN type, although this type accounts in the stomach for only approximately 50%. In contrast to the stomach, other TCGA adenocarcinoma types are rare in the distal oesophagus and GOJ. From this point of view, finding the optimal treatment strategy for oesophageal and GOJ adenocarcinomas could be easier compared to adenocarcinomas of stomach. However, based on the treatment results to date, we can assume that this search will be very complicated and that new active molecules will be needed. The main biomarkers in locally advanced oesophageal and gastro-oesophageal junction adenocarcinomas are listed in Table 2.

## 15. Conclusions

When considering which treatment approach would be most optimal for a patient with distal oesophageal adenocarcinoma or GOJ, currently no molecular predictive marker is suitable. Therefore, we still have to base our decisions on clinical data and results of the randomised studies mentioned in this review. Similarly, worldwide highly recognized guidelines, European Society for Medical Oncology (ESMO) guidelines and National Comprehensive Cancer Network (NCCN) guidelines accept both therapeutic approaches as equivalent, although NCCN slightly prefer preoperative chemoradiotherapy [12,14]. The American Society of Clinical Oncology (ASCO) guideline also accepts both approaches, but the choice is recommended based on clinical parameters (tumour size, lymph node involvement, proximal spread, etc.). For a patient with a large bulky tumour extending proximally, preoperative chemoradiotherapy is recommended to increase the likelihood of R0 resection. Contrarily, for a patient with a small tumour without significant proximal extension and with high probability of curative resection, perioperative chemotherapy would be preferred [13]. Before we know the predictors that will facilitate our decision-making, this recommendation is so far the best guide for choosing a treatment strategy.

## Figures and Tables

**Table 1 cancers-13-04591-t001:** List of the main randomised clinical trials influencing clinical practice in locally advanced oesophageal and gastro-oesophageal junction adenocarcinoma.

Clinical Trial	Design	Number of Patients	Tumour Locations	Results
CROSS	preoperative CRT (RT + paclitaxel + carboplatin) + S vs. S alone	*n* = 366 AC: *n* = 275 (75%)	Proximal oesophagus (2%) Middle oesophagus (13%) Distal oesophagus (58%) GOJ (24%) Missing data (3%)	median OS 49 months vs. 24 months (*p* = 0.003), AC alone: 43 months and 27 months (*p* = 0.038)
SWOG 9008/INT 0116	S + postoperative CRT (RT + 5-FU, LV) vs. S alone	*n* = 556	Cardia (20%) Corpus (24%) Antrum (54%) Multicentric (1%)	median OS 36 months vs. 27 months (*p* = 0.005)
CLASSIC	S + adjuvant capecitabin + oxaliplatin vs. S alone	*n* = 1035	GOJ (2%) Fundus (8%) Fundus and body (2%) Body (33%) Body and antrum (6%) Antrum (46%) Whole gastric (1%) Other (2%)	5-year OS 78% vs. 69% (*p* = 0.0015)
MAGIC	perioperative ECF/ECX + S vs. S alone	*n* = 503	Lower esophagus (14%) GOJ (12%) Stomach (74%)	5-year OS 36% vs. 23% (*p* = 0.009)
FNCLCC/FFCD	perioperative cisplatin + 5FU + S vs. S alone	*n* = 224	Lower esophagus (11%) GOJ (64%) Stomach (25%)	5-year OS 38% vs. 24% (*p* = 0.02)
FLOT4-AIO	Perioperative CT FLOT + S vs. perioperative CT ECF/ECX	*n* = 716	GOJ Siewert type I (23%) GOJ Siewert type II–III (33%) Stomach (44%)	median OS 50 months vs. 35 months (*p* = 0.012)
CheckMate 577	Preoperative CRT + S + adjuvant nivolumab vs. Preoperative CRT + S + placebo	*n* = 794 AC: *n* = 563 (71%)	Oesophagus (60%) GOJ (40%)	median DFS 22.4 months vs. 11.0 months (*p* < 0.001) AC: median DFS 19.4 months vs. 11.1 months

Abbreviations: CRT, chemoradiotherapy; CT, chemotherapy; S, surgery; GOJ, gastro-oesophageal junction; AC, adenocarcinoma; OS, overall survival; DFS, disease-free survival.

**Table 2 cancers-13-04591-t002:** The main predictive biomarkers under research in locally advanced oesophageal and gastro-oesophageal junction adenocarcinomas and current evidence.

Biomarker	Evidence
HER2 expression/*HER2* amplification	The benefit of anti-HER2 therapy has not been demonstrated
PD-L1 expression	The benefit of nivolumab in CheckMate 577 trial was regardless PD-L1 expression. Other studies with immune check-point inhibitors are ongoing
MSI status	MSI status probably influences effect of perioperative chemotherapy could be a predictor of immune check-point inhibitor therapy. Current data are insufficient
EBV status	Current data are insufficient
Tumour mutational burden	Current data are insufficient

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
