# Peer review of "Combined Therapy of Locally Advanced Oesophageal and Gastro–Oesophageal Junction Adenocarcinomas: State of the Art and Aspects of Predictive Factors"

_cancers, 2021, doi:10.3390/cancers13184591_

Round 1
Reviewer 1 Report
In this paper, authors demonstrated an overview of current treatment options in locally advanced adenocarcinomas of oesophagus and GOJ based on the latest evidence. This paper is well written. This manuscript is interesting and clearly presented. This paper includes important points in clinical practice, and I agree with contents of the manuscript. I have one comment on this manuscript.
Comments
- This paper is a complete coverage of chemotherapy and biomarkers for esopgaeal and gastroesophageal junction (GEJ) cancer. For surgeons, please also describe the optimal surgical procedure for esopgaeal and GEJ cancer. Surgery on this site is of interest to surgeons. Because, the site of metastasis of lymph nodes in GEJ is complex.
In patients with Siewert type II and III GEJ cancer with esophageal involvement <4 cm, a prospective nationwide study by the Japanese Gastric Cancer Association and the Japan Esophageal Society suggested perigastric lymphadenectomy and lower mediastinal lymphadenectomy (station 110) as the appropriate removal range [Ann Surg 2021;274:120-7.]. Proximal gastrectomy and lower esophagectomy via transabdominal approach are therefore recommended for these patients. When esophageal involvement is ≥4 cm, however, upper mediastinal lymphadenectomy (station 106) is required, and for these patients we perform McKeown esophagectomy with cervical anastomosis. These two operative procedures are very different.
Author Response
Thanks for the revision of the manuscript and all comments to both reviewers. I am sure that all corrections of the text recommended by the reviewers increased the quality of the article.
Reviewer 1
This paper is a complete coverage of chemotherapy and biomarkers for esophageal and gastroesophageal junction (GEJ) cancer. For surgeons, please also describe the optimal surgical procedure for esopgaeal and GEJ cancer. Surgery on this site is of interest to surgeons. Because, the site of metastasis of lymph nodes in GEJ is complex. In patients with Siewert type II and III GEJ cancer with esophageal involvement <4 cm, a prospective nationwide study by the Japanese Gastric Cancer Association and the Japan Esophageal Society suggested perigastric lymphadenectomy and lower mediastinal lymphadenectomy (station 110) as the appropriate removal range [Ann Surg 2021;274:120-7.]. Proximal gastrectomy and lower esophagectomy via transabdominal approach are therefore recommended for these patients. When esophageal involvement is ≥4 cm, however, upper mediastinal lymphadenectomy (station 106) is required, and for these patients we perform McKeown esophagectomy with cervical anastomosis. These two operative procedures are very different.
A paragraph describing surgical treatment procedures has been added. The surgeon who contributed to this paragraph was added to the co-authors.
Reviewer 2 Report
cancers-1353316
This short review article presents progress towards therapeutic opportunities in gastric adenocarcinoma.
Minor rewording suggestions:
- Introduction: Perhaps the the phrase “clinical research is focused not only on finding a universal optimal treatment approach” should be refined to “clinical research is focused on refining practice guidelines for diagnosis, classification and personalized therapy”.
- Don’t used GOJ and GEJ interchangeably; pick one.
- Clarify that the gene encoding HER2 is ERBB2, and make sure this term complies with gene nomenclature rules by using all capital letters and italics for the gene symbol and no italics for HER2 (ERBB2) protein symbol.
- Remove the word ‘high’ from the phrase ‘worthy to discuss high microsatellite instability’
- Change ‘Ebstein’ to ‘Epstein’; Mention that both EBV status and MSI status impact methylation profiles albeit in different ways, and demethylating treatment strategies are being investigated.
- Add a paragraph about data showing EBV-positive status is associated with response to checkpoint inhibitor Rx.
- Add a paragraph about how inactivated TP53 function (via TP53 gene mutation or deletion) is common in GC and especially in GOJ cancers, as are the downstream effects of TP53 inactivation on factors such as gene copy number changes. Targeted therapy is not available to overcome this mechanism of carcinogenesis.
- Add a paragraph about the ‘genomic stable’ cancers having CDH1 pathway alteration might be susceptible to targeted therapy to restore pathway signalling. These cancers are often recognizable by their distinct histopathologic features.
- Table 2, remove the term ‘yet’; correct grammer in the phrase ‘a could be’; add EBV status to this table.
Author Response
Thanks for the revision of the manuscript and all comments to both reviewers. I am sure that all corrections of the text recommended by the reviewers increased the quality of the article.
Comments Reviewer 2
Introduction: Perhaps the the phrase “clinical research is focused not only on finding a universal optimal treatment approach” should be refined to “clinical research is focused on refining practice guidelines for diagnosis, classification and personalized therapy”. Corrected.
Don’t used GOJ and GEJ interchangeably; pick one. Corrected.
Clarify that the gene encoding HER2 is ERBB2, and make sure this term complies with gene nomenclature rules by using all capital letters and italics for the gene symbol and no italics for HER2 (ERBB2) protein symbol. Corrected.
Remove the word ‘high’ from the phrase ‘worthy to discuss high microsatellite instability’ Corrected
Change ‘Ebstein’ to ‘Epstein’; Corrected
Mention that both EBV status and MSI status impact methylation profiles albeit in different ways, and demethylating treatment strategies are being investigated. Add a paragraph about data showing EBV-positive status is associated with response to checkpoint inhibitor Rx. Corrected.
Add a paragraph about how inactivated TP53 function (via TP53 gene mutation or deletion) is common in GC and especially in GOJ cancers, as are the downstream effects of TP53 inactivation on factors such as gene copy number changes. Targeted therapy is not available to overcome this mechanism of carcinogenesis. Corrected, a paragraph has been added.
Add a paragraph about the ‘genomic stable’ cancers having CDH1 pathway alteration might be susceptible to targeted therapy to restore pathway signalling. These cancers are often recognizable by their distinct histopathologic features. Corrected, a paragraph has been added.
Table 2, remove the term ‘yet’; correct grammer in the phrase ‘a could be’; add EBV status to this table. Corrected.